# Effects of Artificial Intelligence on Surgical Patients’ Health Education

**DOI:** 10.3390/healthcare11202705

**Published:** 2023-10-10

**Authors:** Hsin-Shu Huang, Hsin-Yuan Fang

**Affiliations:** 1Department of Nursing, Central Taiwan University of Science and Technology, Taichung 40601, Taiwan; 2Department of Surgery, China Medical University Hospital, Taichung 40447, Taiwan

**Keywords:** AI automatically generates content, AI image generation tool, National Library e-book storage and reading service

## Abstract

Today, the various abilities that nurses require to meet patients’ healthcare needs adequately are all affected by AI-enabled systems. This research used an experimental study design in which 60 subjects were randomly assigned to either an experimental (AI image e-book guidance) group or a control (text paper guidance) group after meeting the admission conditions and agreeing to participate in the study. It was proven that providing AI image e-book guidance before surgery significantly changed the behavior of patients and promoted relief of urinary catheter discomfort through self-efficacy to reduce urinary catheter pain after surgery (*p* < 0.001). It was found that providing AI image e-book guidance can shorten the time for health education and provide patients with repeated medical education and familiarity with health guidance, which can help to address the important clinical service demand issue and the shortage of nursing staff.

## 1. Introduction

The literature indicates that the prevalence of catheter-related bladder discomfort (CRBD) after urological surgery is about “47–84.5%” [1,2], and the prevalence after discectomy is 62.9% [3]. Common symptoms of discectomy are sciatica, weakness, and numbness in the lower limbs. More seriously, cauda equina syndrome, which can affect urinary and defecatory functions, has a 0.2–1% combined occurrence after surgery [4]. As many as 47–91% of patients who regain consciousness in the recovery room after anesthesia experience CRBD caused by the placement of urinary catheters during surgery [5]. The symptoms of CRBD can lead postsurgery patients to experience distress and agitated emotions, even to the point of self-removal of the catheter because they cannot bear the discomfort it causes [6]. Moreover, it can lead to the intensification of post-surgery pain or unstable vital signs, ultimately affecting postsurgery recovery and increasing days of hospitalization. Considering the importance of patient safety, mutual dedication to the effective health education of patients with urinary catheters is a pressing and required matter.

Images and words are easier to learn from than words alone, which in psychology is called the “image effect” [7]. Images are often used to assist the understanding and learning of words. The use of rich and diverse images in health and education resources can provide concise and clear visual communication to improve memory so that the teaching content can be easily stored in the right brain, and learning through puzzles can often strengthen learners’ knowledge and skills [8].

Health education uses complementary teaching to provide or improve patient health knowledge, develop healthy attitudes, construct healthy behaviors, or change behaviors to promote health through cognitive, situational, and skill-level interventions, with the aim of teaching patients the skills and knowledge that they require to take care of themselves. Health education is mainly used to help patients achieve enhanced self-care to improve their quality of life [9]. In view of the short time available for a patient’s pre-operative education, text paper guidance content is not a simple and clear option; pictures and images can promote the deepening of the patient’s memory, enhance comprehension and attention as well as learners’ interest in reading, and clearly express the health content [10]. 

Scholars believe that patients’ awareness of their own diseases is related to self-care behavior, and health education guidance can improve patients’ disease awareness and effectively improve their self-care ability [10]. In addition, studies have shown that health education has a positive impact on patient cognition and that self-efficacy enables patients to take action after obtaining relevant knowledge from healthcare professionals [11]. The construction of self-efficacy is the inner process of self-persuasion, which is a complex cognitive process. People with higher self-efficacy will continue to work hard, accept, and overcome, even when encountering difficulties; thus, enhancing personal self-efficacy will lead to behavioral changes. Therefore, if patients have a good understanding of themselves, have self-confidence, and learn breathing relaxation skills, it can help to alleviate the bladder discomfort caused by urinary catheterization and encourage them to participate in self-care motivation, implement healthy self-management in life, and strive for their own health and quality of life [12].

Medicine serves common human needs, such as promoting patient well-being and making adequate health care available to all. Meanwhile, we have a good picture of what patients want and need with regard to their own care, making sure that it is patients who benefit the most from the surge of AI health technology [13].

This study investigated the effectiveness of AI image e-book guidance and text paper guidance to improve CRBD, patients’ self-efficacy to improve discomfort caused by urinary catheters, the level of pain caused by urinary catheters, and the severity of CRBD. Through AI image e-book guidance, the learners’ ability to solve health-related problems and to improve behaviors of self-care was strengthened, thus effectively reducing the discomfort caused by catheterization. 

## 2. Materials and Methods

### 2.1. Study Design and Participants

In this experimental research design, study participants were randomly selected using MS Excel 2019 from the list of all cases from a certain hospital that fit the criteria. Those who agreed to receive AI image e-book guidance were randomly assigned to the experimental and control groups using MS Excel 2019. Pretests were completed by both groups prior to AI image e-book guidance. The experimental group members received AI image e-book guidance that was provided to reduce urinary catheter discomfort, as well as the text paper guidance received by the control group. Both groups completed a posttest 15 min after the intervention, and both groups completed a questionnaire assessing self-efficacy to improve bladder discomfort caused by the urinary catheter, the scale of CRBD severity, and the level of pain caused by the urinary catheter in order to collect pretest and posttest data. In order to compare the differences in scores on the scale of self-efficacy to improve discomfort caused by the urinary catheter, severity of CRBD, and the level of pain caused by the urinary catheter before AI image e-book guidance, 15 min after intervention, 6 h after surgery, and 24 h after surgery, independent-sample *t*-tests and repeated measures ANOVA were conducted.

The researcher obtained the data on patients using urinary catheters during orthopedic or neurospinal surgery patients from a certain medical center in central Taiwan. Those subjects who met the collection criteria and agreed to participate in the research were incorporated into the sample pool. The cases were numbered, and MS Excel 2019 was used for random number sampling to group subjects into experimental and control groups. The research sample estimation method uses G-Power version 3.1 software calculation and uses ANOVA to set effect size = 0.25 (medium), alpha level = 0.05, and power = 0.8. The calculation result is based on a sample number of 60 subjects. In the estimated sample collection, there are 30 subjects in each of the experimental and control groups, making a total of 60 subjects. Basic information, such as age, level of education, marital status, etc., of the two groups did not have significant differences when examined using a chi-square test, *p* > 0.05. Variables, including the scale of self-efficacy to improve discomfort caused by urinary catheterization, level of pain caused by urinary catheters, and severity of catheter-related bladder discomfort of the two groups, did not have significant differences when examined using independent *t*-tests, *p* > 0.05; thus, the two groups had similar conditions prior to intervention.

### 2.2. Measurement

#### 2.2.1. AI Image E-Book Guidance to Improve Urinary Catheter Discomfort

The design of the AI image e-book guidance to improve urinary catheter discomfort referred to previous literature related to discomfort caused by catheters after surgery and the characteristics of the research subjects [14]. The whole AI image e-book guidance process took approximately 10–15 min, and the process was as follows: before the surgery, patients were allowed to understand the purpose, function, and placement time of the urinary catheter through AI image e-book guidance, and the placement process and principle of urine drainage were explained. Catheter-related discomfort and symptoms after awakening from anesthetics were also explained, and methods were provided to relieve the discomfort by diverting attention (listening to music, reading books, watching TV, and taking comfortable lying positions). The nursing staff took a one-on-one approach to teaching patients, using the AI image e-book guidance to allow patients to watch and understand breathing and relaxation techniques to relieve catheter discomfort. The patients were shown breathing and relaxation techniques that they could utilize whenever they felt discomfort from the catheter. 

The AI image e-book guidance was assessed for expert validity. In the AI image e-book guidance design process, the researcher invited five expert scholars in related fields: a urology attending physician (assistant professor in the School of Medicine), another urology attending physician, two urology nurse practitioners, and the head nurse of the urology ward. The five experts provided improvement suggestions on the content of the AI image e-book, ensuring that the AI image e-book possessed good expert validity. The expert content validation assessed the usability, clarity, and suitability of the scale questions. The researcher used the index of content validity (CVI) to assess the content of the AI image e-book. The content validity index CVI value is the expert validity (expert rating index). The calculation method of the expert rating index is the ratio of the number of experts applicable to the assessment divided by the number of experts who scored all the ratings. The researcher calculated the average score of the usability, clarity, and suitability of each assessment, and the results showed that each assessment had a CVI between 0.86 and 1.

#### 2.2.2. Scale of Self-Efficacy to Improve Discomfort Caused by Urinary Catheterization 

The scale of self-efficacy to improve discomfort caused by urinary catheterization was taken from the “General Self-Efficacy Scale (GSE)”, which is public and can be freely used by researchers [15]. The researcher added the words “improve discomfort caused by urinary catheter” to each question based on the current clinical situation and with the premise of not changing the original meaning of the question. The CVI assessment was conducted based on necessity, suitability, and importance, and the CVI values were mostly between 0.88 and 0.93; in terms of reliability, the Cronbach’s value was 0.83–0.93; a standard Likert scoring method of a 5-point scale was used, and scores were graded between 1 and 5. The lowest total score was 10, and the highest was 50; the higher the summed score, the better the self-efficacy for the improvement of discomfort caused by the catheter.

#### 2.2.3. Level of Pain Caused by Urinary Catheters

The pain visual analog scale (VAS) is a continuous straight line, where the left- and right-most sides of the 10 cm line represent the level of pain. A score of 0 represents “no pain at all”, and 10 represents “unbearable pain”. Patients can give scores indicating their self-perceptions of pain and discomfort [16]. In terms of the construct validity of the NRS, results showed that the NRS and VAS had a high correlation (r = 0.847, *p* < 0.001). On this scale, 1–3 points represent mild pain, 4–6 points represent medium pain, and 7–10 point represents severe pain; the higher the score, the greater the discomfort and pain caused by the urinary catheter [17].

#### 2.2.4. Severity of Catheter-Related Bladder Discomfort

The severity of catheter-related bladder discomfort scale did not previously have a Chinese version; thus, the scale was translated. The CVI was analyzed on the basis of necessity, suitability, and importance, with the results mostly between 0.97 and 1; the Cronbach’s value of reliability was 0.88–0.90 [2,3,18]. A standard scoring method involving a Likert 4-point scale was used. Scores were graded between 1 and 4, with 1 being the lowest and 4 being the highest; 1 point represents no discomfort, 2 points mild discomfort, 3 points medium discomfort, and 4 points severe discomfort. The higher the score, the greater the severity of catheter-related bladder discomfort.

### 2.3. Research Time and Place

This research required 48 h between 17 February 2022 and 9 May 2022. The researcher chose to start the test from the day before the surgery and after completing the admission process so that the activities were not interfered with by others. The experimental group received AI image e-book guidance for 15–20 min (including relaxation techniques and repeated demonstrations). The pretest was conducted before image e-book guidance was provided, and the posttest timings were 15 min after image e-book guidance, 6 h after surgery, and 24 h after surgery. The control group received text paper guidance for 15–20 min (including relaxation techniques and repeated demonstrations). The pretest was conducted before text guidance was provided, and the post-test timings were 15 min after text guidance, 6 h after surgery, and 24 h after surgery.

### 2.4. Ethical Considerations

This research project was reviewed and approved by the IRB of China Medical University Hospital (No: CMUH110-REC3-211). The patients could quit at any time and had the right to raise questions; the questionnaires were anonymous. The questionnaire results were numbered anonymously to delink and ensure confidentiality. The patients’ names and conditions will never be publicized, and the results are for academic use only.

### 2.5. Data Collection and Analysis 

This study took 450 orthopedic and neurospinal surgery patients from a certain medical center in central Taiwan as the research population. The collection criteria were patients who could communicate in Mandarin or Taiwanese, had been diagnosed by a physician to have herniated intervertebral discs in the lumbar area, and had been admitted to the hospital for lumbar vertebral surgery. The exclusion criteria were the following: those who were unconscious and could not communicate; those who did not wish to participate; those with cystitis, frequent urination, difficulty urinating or neurogenic bladder, terminal kidney disease (urine volume less than 500 mL/24 h), coagulopathy, overactive bladder, and long-term usage of painkillers. The data collection process randomly assigned cases that fit the criteria into experimental and control groups, confirmed the research subjects, and acquired IRB consent. The data collection personnel collected data for the researcher via questionnaire; the collection tools were the scales, and the questionnaire structure was close-ended (self-efficacy to improve discomfort caused by the urinary catheter, level of pain caused by the urinary catheter, and severity of CRBD). The method of data analysis was confirmed, an oral explanation was given to the patients, and then the scaled questionnaires were distributed in a public setting. The collected data were examined, and the pretest and posttest data were formally collected.

SPSS 25.0 for Windows Mandarin version was used for statistical analysis, and the independent *t*-test and chi-square test were conducted to compare the homogeneity of the experimental and control groups. For assessment of the effectiveness of the AI image e-book guidance in improving catheter discomfort, analyses involving independent *t*-tests and repeated measures ANOVA were conducted to analyze the differences according to variables of the experimental and control groups at different times.

## 3. Results

### 3.1. Homogeneity Examination of Each Index in Both Groups Prior to Intervention

The basic information of both groups was analyzed using the chi-square test and *t*-test to examine homogeneity; the results indicated no significant difference (*p* > 0.05). Therefore, this showed that the two groups were comparable.

### 3.2. Analyzing the Amount of Change during the Pre- and Posttests of Both Groups in Self-Efficacy to Improve the Discomfort Caused by the Urinary Catheter, Level of Pain Caused by the Urinary Catheter, and Severity of CRBD

#### 3.2.1. Comparative Analysis of Experimental and Control Groups Regarding Self-Efficacy to Improve the Discomfort Caused by the Urinary Catheter

The pretest of the experimental group on self-efficacy to improve the discomfort caused by the urinary catheter (the catheter was not placed prior to the health education intervention) produced an average score of 1.21, and the third posttest score (24 h after surgery, catheter placed) averaged 4.87. The average amount of change in the score was 3.66 points; the pretest of the control group on self-efficacy to improve the discomfort caused by the urinary catheter (the catheter was not placed prior to the health education intervention) produced an average score of 1.12, and the third posttest score (24 h after surgery, catheter placed) averaged 1.17. The average amount of change in the score was 0.05 points; the amount of change in the score between the pretest and third posttest of self-efficacy to improve the discomfort caused by the urinary catheter in the two groups underwent independent *t*-test analysis, which produced a result of *t* = 33.560, *p* = 0.000. The two groups had significant differences in the average score change between the pretest and the third posttest, indicating that AI image e-book guidance improved patients’ self-efficacy to improve the discomfort caused by their catheters (Table 1).

#### 3.2.2. Level of Pain Caused by Urinary Catheter 

The pretest of the experimental group on the level of pain caused by the catheter (the catheter was not placed prior to the health education intervention) produced an average score of 0, and the third posttest score (24 h after surgery, catheter placed) averaged 0.1. The average amount of change in the score was 0.1 points. The pretest of the control group on the level of pain caused by the catheter (the catheter was not placed prior to the health education intervention) produced an average score of 0, and the third posttest score (24 h after surgery, catheter placed) averaged 5.13. The average amount of change in the score was 5.13 points; the amount of change in the score between the pretest and third posttest of the level of pain caused by the catheter in the two groups underwent independent *t*-test analysis, which produced a result of *t* = 36.924, *p* < 0.001, indicating that the AI image e-book guidance reduced the level of pain caused by the catheter (Table 1).

#### 3.2.3. Severity of Catheter-Related Bladder Discomfort

The pretest of the experimental group on the severity of CRBD (the catheter was not placed prior to the health education intervention) produced an average score of 0, and the third posttest score (24 h after surgery, catheter placed) averaged 0.07. The average amount of change in the score was 0.07 points; the pretest of the control group on the severity of CRBD (the catheter was not placed prior to the health education) had an average score of 0, and the third posttest score (24 h after surgery, catheter placed) averaged 2.97. The average amount of change in the score was 2.97 points; the amount of change in the score between the pretest and third posttest of the severity of CRBD in the two groups underwent an independent *t*-test, which produced a result of *t* = 22.209, *p* < 0.001, indicating that AI image e-book guidance reduced the severity of catheter-related bladder discomfort (Table 1).

After the AI image e-book guidance for improvement of catheter-related bladder discomfort, the self-efficacy to improve the discomfort caused by the urinary catheter in the experimental group significantly increased. Its utilization was associated with significant reductions in pain and discomfort compared to the control group in terms of the level of pain caused by the urinary catheter and the severity of CRBD (Table 1).

### 3.3. Analysis of Score Change between Pre- and Posttests of Both Groups on the Self-Efficacy to Improve the Discomfort Caused by the Urinary Catheter, Level of Pain Caused by the Urinary Catheter, and Severity of CRBD 

#### 3.3.1. Self-Efficacy to Improve the Discomfort Caused by the Urinary Catheter

Regarding the self-efficacy to improve the discomfort caused by the catheter, the average scores of the pretest and posttest 15 min after the health education intervention in the experimental group were 1.21 and 3.28, respectively; the 2.06 difference in average value was significant (*p* < 0.001). The average scores of the posttests 15 min after the health education intervention (catheter not yet placed) and 6 h after surgery (catheter already placed) were 3.28 and 4.42, respectively; the difference in average value was 1.14 (*p* < 0.001). Six hours after surgery (catheter already placed) and 24 h after surgery (catheter already placed), the average scores were 4.42 and 4.87, respectively, with the difference in average value being 0.44 (*p* < 0.001). The average scores of the pretest and posttest 24 h after surgery (catheter already placed) were 1.21 and 4.87, respectively, with the difference in average value being 3.65 (*p* < 0.001). The above results indicate that self-efficacy to improve the discomfort caused by the catheter was significantly improved after AI image e-book guidance was provided (Table 2).

Regarding the self-efficacy to improve the discomfort caused by the catheter, the average scores of the pretest and posttest at 15 min after the health education intervention among the control group were 1.12 and 1.15, respectively; the 0.03 difference in the average value was not significant (*p* = 0.533). The average scores of the pos-tests 15 min after health education (catheter not yet placed) and 6 h after surgery (catheter already placed) were 1.15 and 1.14, respectively, with the difference in average value being −0.01 (*p* = 0.654). Six hours after surgery (catheter already placed) and 24 h after surgery (catheter already placed), the average scores were 1.14 and 1.17, respectively, with the difference in average value being 0.03 (*p* = 0.202). The average scores of the pretest and posttest at 24 h after surgery (catheter already placed) were 1.12 and 1.17, respectively, and the difference in average value was 0.05 (*p* = 0.222). The above results, with no significant differences, indicate that self-efficacy to improve the discomfort caused by the catheter was not improved after text paper guidance was provided for the improvement of catheter discomfort (Table 3).

#### 3.3.2. Level of Pain Caused by Urinary Catheter

Regarding the level of pain caused by the catheter, the catheter was not placed at the time of the pretest and the posttest 15 min after the health education intervention. The score for the experimental group was 0 for both tests; the average scores of the posttests 15 min after health education (catheter not yet placed) and 6 h after surgery (catheter already placed) were 0 and 0.60, respectively, with the difference in average value being 0.60 (*p* < 0.001). Six hours after surgery (catheter already placed) and 24 h after surgery (catheter already placed), the average scores were 0.60 and 0.10, respectively, with the difference in average value being −0.50 (*p* < 0.001). The average scores of the pretest (catheter not yet placed) and posttest 24 h after surgery (catheter already placed) were 0 and 0.10, respectively, and the difference in average value was 0.10 (*p* = 0.184). The above results indicate that the level of pain caused by the catheter was significantly improved after AI image e-book guidance was provided for the improvement of catheter discomfort (Table 4).

In terms of the level of pain caused by the catheter, the catheter was not placed at the time of the pretest and the posttest 15 min after the health education intervention; the score of the control group was 0 for both tests. The average scores of posttests 15 min after health education (catheter not yet placed) and 6 h after surgery (catheter already placed) were 0 and 5.00, respectively, with the difference in average value being 5.00 (*p* < 0.001). Six hours after surgery (catheter already placed) and 24 h after surgery (catheter already placed), the average scores were 5.00 and 5.13, respectively, and the difference in average value was 0.13 (*p* = 0.595). The average scores of the pretest and posttest 24 h after surgery (catheter already placed) were 0 and 5.13, respectively, with the difference in average value being 5.13 (*p* < 0.001). The above results show significant differences, and the average score for the level of pain caused by the catheter after patients received text paper guidance alone was higher than that for the experimental group (Table 5).

#### 3.3.3. Severity of Catheter-Related Bladder Discomfort

Regarding the severity of CRBD, the catheter was not placed at the time of the pretest and the posttest 15 min after the health education intervention; the score of the experimental group was 0 for both tests. The average scores of the posttests 15 min after the health education intervention (catheter not yet placed) and 6 h after surgery (catheter already placed) were 0 and 0.43, respectively, and the difference in average value was 0.43 (*p* < 0.001). Six hours after surgery (catheter already placed) and 24 h after surgery (catheter already placed) the average scores were 0.43 and 0.07, respectively, with the difference in average value being −0.36 (*p* < 0.001). The average scores of the pretest and posttest 24 h after surgery (catheter already placed) were 0 and 0.07, respectively, with the difference in average value being 0.067 (*p* = 0.161). The above results indicate that the severity of CRBD significantly improved after AI image e-book guidance was provided for the improvement of catheter discomfort. The average score difference between the pre-test (catheter not yet placed) and 24 h after surgery was 0.067 (*p* = 0.161), indicating that the severity of catheter-related bladder discomfort was the same as when the catheter was not yet placed (Table 6).

In terms of the severity of CRBD, the catheter was not placed at the time of the pretest and the posttest 15 min after the health education intervention; the score for the control group was 0 for both tests. The average scores of the posttests 15 min after health education (catheter not yet placed) and 6 h after surgery (catheter already placed) were 0 and 2.8, respectively, with the difference in average value being 2.8 (*p* < 0.001). Six hours after surgery (catheter already placed) and 24 h after surgery (catheter already placed) had average scores of 2.8 and 2.97, respectively, with the difference in average value being 0.167 (*p* = 0.444). The average scores of the pretest (catheter not yet placed) and posttest 24 h after surgery (catheter already placed) were 0 and 2.97, respectively, and the difference in average value was 2.97 (*p* = 0.161). The above results indicate that the average score of the severity of CRBD after the health education intervention was provided was higher than the experimental group, and the intervention, therefore, did not significantly reduce the severity of CRBD (Table 7).

## 4. Discussion

The AI image e-book guidance provided for the improvement of the discomfort caused by urinary catheterization resulted in significant improvement in the patients’ self-efficacy to improve discomfort caused by the catheter, the level of pain caused by the catheter, and the severity of CRBD. During the AI image e-book guidance process in this research, it was found that patients could focus on the AI image e-book guidance and understand and remember its content quickly. Patient familiarity with and correctness of relaxation techniques were improved when the techniques were demonstrated repeatedly. It is suggested that AI image e-book guidance for improvement of the discomfort caused by urinary catheterization can be incorporated into presurgery preparation in order to benefit patients’ understanding of the related purpose and principles of urinary catheters before they are placed. The relaxation techniques can also be utilized immediately after surgery to improve the discomfort caused by the catheter.

This research proves that AI image e-book guidance provided for the improvement of discomfort caused by urinary catheters can significantly improve patients’ self-efficacy to improve the discomfort caused by urinary catheters, the level of pain caused by urinary catheters, and the severity of CRBD. If data from multiple departments can be compared and analyzed, there will be more departments able to clarify the requirements of the intervention. This is the limitation of the study; hence, this work can be combined with work conducted in other departments or medical centers for expanded sample collection in the future in order to continue and deepen the research.

## 5. Conclusions

This research discovered that providing AI image e-book guidance can reduce the time required for repeated health education to patients to familiarize them with instructions. Providing AI image e-book guidance for the improvement of discomfort caused by urinary catheters prior to surgery can assist patients in learning and enhancing their self-efficacy to alleviate the discomfort caused by catheters and relieve the discomfort and pain caused by catheters after surgery. This can help to address the important clinical service demand issue and shortage of nursing staff.

## Figures and Tables

**Table 1 healthcare-11-02705-t001:** Average scores and their differences for the three scales before and after the AI image e-book guidance for improvement of the discomfort caused by the urinary catheter was provided: self-efficacy to improve discomfort caused by urinary catheter, level of pain caused by urinary catheter, and severity of CRBD (N = 60).

Type	Group	Average		Average of Change in Score	*t*	*p*	Average		Average of Change in Score	*t*	*p*	Average		Average of Change in Score	*t*	*p*
		Pretest (Catheter Not yet Placed)	First Posttest (Catheter Not yet Placed)				Pretest (Catheter Not yet Placed)	Second Posttest (Catheter Placed)				Pretest (Catheter Not yet Placed)	Third Posttest (Catheter Placed)			
Self-efficacy to improve discomfort caused by urinary catheter	Experimental Group	1.21	3.28	2.07	12.443	0	1.21	4.42	3.21	24.812	0	1.21	4.87	3.66	33.56	0
	Control Group	1.12	1.15	0.03			1.12	1.14	0.02			1.12	1.17	0.05		
Level of pain caused by urinary catheter	Experimental Group	0	0	0	-	-	0	0.6	0.6	18.137	0	0	0.1	0.1	36.924	0
	Control Group	0	0	0			0	5	5			0	5.13	5.13		
Severity of CRBD	Experimental Group	0	0	0	-	-	0	0.43	0.43	12.308	0	0	0.07	0.07	22.209	0
	Control Group	0	0	0			0	2.8	2.8			0	2.97	2.97		

**Table 2 healthcare-11-02705-t002:** Repeated measures ANOVA of self-efficacy to improve the discomfort caused by the urinary catheter in experimental group (N = 30).

Timing (Visit)	Self-Efficacy to Improve Discomfort Caused by Catheter			
	Mean (SD)	Difference in Average	*p*-Value ^a^	*p*-Value ^b^
Pretest (Catheter not yet placed)	1.21 (0.279)	3.65	0.000	-
15 min after health education (Catheter not yet placed)	3.28 (0.724)	2.06		0.000
6 h after surgery (Catheter placed)	4.42 (0.662)	1.14		0.000
24 h after surgery (Catheter placed)	4.87 (0.468)	0.44		0.000

^a^ Paired-sample *t*-test. ^b^ Repeated measures ANOVA.

**Table 3 healthcare-11-02705-t003:** Repeated measures ANOVA of self-efficacy to improve the discomfort caused by the urinary catheter in the control group (N = 30).

Timing (Visit)	Self-Efficacy to Improve Discomfort Caused by Catheter			
	Mean (SD)	Difference in Average	*p*-Value ^a^	*p*-Value ^b^
Pretest (Catheter not yet placed)	1.12 (0.207)	0.050	0.222	-
15 min after health education (Catheter not yet placed)	1.15 (0.140)	0.030		0.533
6 h after surgery (Catheter placed)	1.14 (0.894)	−0.013		0.654
24 h after surgery (Catheter placed)	1.17 (0.114)	0.033		0.202

^a^ Paired-sample *t*-test. ^b^ Repeated measures ANOVA.

**Table 4 healthcare-11-02705-t004:** Repeated measures ANOVA of the level of pain caused by urinary catheter in the experimental group (N = 30).

Timing (Visit)	Level of Pain Caused by Catheter			
	Mean (SD)	Difference in Average	*p*-Value ^a^	*p*-Value ^b^
Pretest (Catheter not yet placed)	0.00 (0.000)	0.100	0.184	-
15 min after health education (Catheter not yet placed)	0.00 (0.000)	0.000		-
6 h after surgery (Catheter placed)	0.60 (0.814)	0.600		0.000
24 h after surgery (Catheter placed)	0.10 (0.403)	−0.500		0.000

^a^ Paired-sample *t*-test. ^b^ Repeated measures ANOVA.

**Table 5 healthcare-11-02705-t005:** Repeated measures ANOVA of the level of pain caused by urinary catheter in the control group (N = 30).

Timing (Visit)	Level of Pain Caused by Catheter			
	Mean (SD)	Difference in Average	*p*-Value ^a^	*p*-Value ^b^
Pretest (Catheter not yet placed)	0.00 (0.00)	5.13	0.000	-
15 min after health education (Catheter not yet placed)	0.00 (0.00)	0.00		-
6 h after surgery (Catheter placed)	5.00 (1.050)	5.00		0.000
24 h after surgery (Catheter placed)	5.13 (0.629)	0.13		0.595

^a^ Paired-sample *t*-test. ^b^ Repeated measures ANOVA.

**Table 6 healthcare-11-02705-t006:** Repeated measures ANOVA of the severity of CRBD in the experimental group (N = 30).

Timing (Visit)	Severity of CRBD			
	Mean (SD)	Difference in Average	*p*-Value ^a^	*p*-Value ^b^
Pretest (Catheter not yet placed)	0.000 (0.000)	0.067	0.161	-
15 min after health education (Catheter not yet placed)	0.000 (0.000)	0.000		-
6 h after surgery (Catheter placed)	0.433 (0.568)	0.433		0.000
24 h after surgery (Catheter placed)	0.07 (0.254)	−0.367		0.000

^a^ Paired-sample *t*-test. ^b^ Repeated measures ANOVA.

**Table 7 healthcare-11-02705-t007:** Repeated measures ANOVA of the severity of CRBD in the control group (N = 30).

Timing (Visit)	Severity of CRBD			
	Mean (SD)	Difference in Average	*p*-Value ^a^	*p*-Value ^b^
Pretest (Catheter not yet placed)	0.00 (0.00)	2.967	0.000	-
15 min after health education (Catheter not yet placed)	0.00 (0.00)	0.000		-
6 h after surgery (Catheter placed)	2.80 (0.88)	2.800		0.000
24 h after surgery (Catheter placed)	2.97 (0.66)	0.167		0.444

^a^ Paired-sample *t*-test. ^b^ Repeated measures ANOVA.

## Data Availability

All of the relevant datasets in this study are described in the manuscript.

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
