# Peer review of "Effects of Artificial Intelligence on Surgical Patients’ Health Education"

_healthcare, 2023, doi:10.3390/healthcare11202705_

Round 1
Reviewer 1 Report
The manuscript aims to present the result of how the use of IA image e-book guidance can improve the relief of urinary catheter discomfort versus its paper-based counterparty, before surgery.
The abstract accurately represents the manuscript, but it would be useful if you mention in this section what were the circumstances and test that produced the p=0.000 value. Also, it would be beneficial for the paper in you mention what techniques of IA are included in the AI-image e-book.
The keyword could perhaps include the techniques of IA used in AI image e-book due to AI is very wide term.
The introduction section is adequate and gives a concise idea of what is the objective of the study. In this section authors could also give a glance about the design of the e-book.
Materials and Methods section is also adequate and gives a very good description of the parts and sequence of the study.
Results section is adequate and show the findings of the realized statistical tests.
In conclusion section, although all statistical tests are important, the authors could give an opinion on which they consider the most important to support their conclusions.
No comments
Author Response
Response to Reviewer Comments
Comments and Suggestions for Authors
The manuscript aims to present the result of how the use of IA image e-book guidance can improve the relief of urinary catheter discomfort versus its paper-based counterparty, before surgery.
The abstract accurately represents the manuscript, but it would be useful if you mention in this section what were the circumstances and test that produced the p=0.000 value. Also, it would be beneficial for the paper in you mention what techniques of IA are included in the AI-image e-book.
The keyword could perhaps include the techniques of IA used in AI image e-book due to AI is very wide term.
The introduction section is adequate and gives a concise idea of what is the objective of the study. In this section authors could also give a glance about the design of the e-book.
Materials and Methods section is also adequate and gives a very good description of the parts and sequence of the study.
Results section is adequate and show the findings of the realized statistical tests.
In conclusion section, although all statistical tests are important, the authors could give an opinion on which they consider the most important to support their conclusions.
Comments on the Quality of English Language
No comments
Response:
Thanks to the reviewer for his insightful suggestions, the authors have revised the keywords in the attached document line 18-19 to AI automatically generates content; AI image generation tool; National Library e-book storage and reading service.

Reviewer 2 Report
The authors present the effectiveness of AI image e-book guidance and text paper guidance to improve CRBD, patients’ self-efficacy to improve discomfort caused by urinary catheters, the level of pain caused by urinary catheters, and the severity of CRBD. I read the manuscript with great interest and believe its topic is important and relevant. There are some comments aimed at improving the quality of the paper:
1. The authors should spend more time describing the outline of AI image e-book guidance and the features of the presented literature.
2. It is suggested that this article add a section of “CRBD on the current status of nursing care in your research field” in the paper.
3. It is suggested that this article also add a section on “CRBD on the assessment scale” in the paper.
4. Do they all have the same surgery and Indwelling urinary catheter method for the 60 reached subjects?
5. Reference tables should according to the order mentioned in the text in order of appearance, and close to content.
Author Response
Response to Reviewer Comments
Comments and Suggestions for Authors
The authors present the effectiveness of AI image e-book guidance and text paper guidance to improve CRBD, patients’ self-efficacy to improve discomfort caused by urinary catheters, the level of pain caused by urinary catheters, and the severity of CRBD. I read the manuscript with great interest and believe its topic is important and relevant. There are some comments aimed at improving the quality of the paper:
- The authors should spend more time describing the outline of AI image e-book guidance and the features of the presented literature.
- It is suggested that this article add a section of “CRBD on the current status of nursing care in your research field” in the paper.
- It is suggested that this article also add a section on “CRBD on the assessment scale” in the paper.
- Do they all have the same surgery and Indwelling urinary catheter method for the 60 reached subjects?
- Reference tables should according to the order mentioned in the text in order of appearance, and close to content.
Response:
Thanks to the reviewer for his insightful suggestions, the authors explain the conditions of the research subjects in the attached document lines 91-94.

Reviewer 3 Report
Manuscript ID: healthcare-2620554
Title: Effects of Artificial Intelligence on Surgical Patients’ Health Education
This manuscript presents a study to claim that providing “AI image e-book” guidance before orthopedic and neuro-spinal surgery has significantly changed the behavior of patients and promoted relief of urinary catheter discomfort through self-efficacy to reduce urinary catheter pain. However, the title has the term ‘artificial intelligence,’ there is no discussion of artificial intelligence method/ techniques in the paper, except AI image e-book in which AI is also not explicitly mentioned as artificial intelligence, at least once.
The paper is well written and presented the results, I’ve following comments:
1. How do authors justify the term artificial intelligence in the title? What is the AI image e-book and how do researchers use it? There is no use of AI/ ML techniques in data processing or analysis. Data were processed using statistical techniques with SPSS tool. Authors should clearly explain the suitability of the term artificial intelligence for this paper.
2. The introduction section should be improved by discussing other relevant research studies. There should be some details of the AI image e-book or at least some brief information.
3. Initially, responses were recorded through questionnaire (in Mandarin/ Taiwanese), and scores were calculated considering Chinese scale (1~4) and then scores translated to 1~10 scale. Wouldn’t it affect the accuracy of actual response for the level of pain? How do authors have overcome this issue?
4. Page 2, line 76~79: “In this experimental research design, study participants were randomly selected using Excel…” Here meaning of the sentence is not very clear; also, please mention whether it is MS-Excel or something else.
5. Page 2, line 82~86: “Both groups completed…. , in order to collect pre-test and post-test data”: this sentence should be restructured for clear meaning.
6. What is index of content validity (CVI), how it is defined and calculated?
7. Why have subjects/ participants chosen randomly? There should be some participant inclusion criterion in such cohort study.
8. Page 4, Line 195: “SPSS for Windows 25.0…” I think it should be SPSS 25.0 for Windows.
9. Responses were recorded from 450 orthopedic and neuro-spinal surgery patients for 48 hours of study. Then, how it become 60 data samples from 30 subjects? Is there some data exclusion criterion as well? Authors are advised to include a figure depicting the total number of subjects participated, data inclusion criterion, number of subjects whose data considered, etc.
10. In Table 7, please keep enough spaces between table column headings to avoid confusion. Here, experimental group for first post-test (catheter not yet placed) t is 12.443, and for control group with second post-test (catheter placed) t is 12.308. How to justify these two very close values for different settings?
11. Authors are advised to restructure the long sentences into smaller ones with clear meaning.
Overall, the paper contains interesting and revealing content. Authors are advised for minor revision of this paper.
Authors are advised to restructure the long sentences into smaller ones with clear meaning to avoid any ambiguity.
Author Response
Response to Reviewer Comments
Comments and Suggestions for Authors
Manuscript ID: healthcare-2620554
Title: Effects of Artificial Intelligence on Surgical Patients’ Health Education
This manuscript presents a study to claim that providing “AI image e-book” guidance before orthopedic and neuro-spinal surgery has significantly changed the behavior of patients and promoted relief of urinary catheter discomfort through self-efficacy to reduce urinary catheter pain. However, the title has the term ‘artificial intelligence,’ there is no discussion of artificial intelligence method/ techniques in the paper, except AI image e-book in which AI is also not explicitly mentioned as artificial intelligence, at least once.
The paper is well written and presented the results, I’ve following comments:
- How do authors justify the term artificial intelligence in the title? What is the AI image e-book and how do researchers use it? There is no use of AI/ ML techniques in data processing or analysis. Data were processed using statistical techniques with SPSS tool. Authors should clearly explain the suitability of the term artificial intelligence for this paper.
- The introduction section should be improved by discussing other relevant research studies. There should be some details of the AI image e-book or at least some brief information.
- Initially, responses were recorded through questionnaire (in Mandarin/ Taiwanese), and scores were calculated considering Chinese scale (1~4) and then scores translated to 1~10 scale. Wouldn’t it affect the accuracy of actual response for the level of pain? How do authors have overcome this issue?
- Page 2, line 76~79: “In this experimental research design, study participants were randomly selected using Excel…” Here meaning of the sentence is not very clear; also, please mention whether it is MS-Excel or something else.
- Page 2, line 82~86: “Both groups completed…. , in order to collect pre-test and post-test data”: this sentence should be restructured for clear meaning.
- What is index of content validity (CVI), how it is defined and calculated?
- Why have subjects/ participants chosen randomly? There should be some participant inclusion criterion in such cohort study.
- Page 4, Line 195: “SPSS for Windows 25.0…” I think it should be SPSS 25.0 for Windows.
- Responses were recorded from 450 orthopedic and neuro-spinal surgery patients for 48 hours of study. Then, how it become 60 data samples from 30 subjects? Is there some data exclusion criterion as well? Authors are advised to include a figure depicting the total number of subjects participated, data inclusion criterion, number of subjects whose data considered, etc.
- In Table 7, please keep enough spaces between table column headings to avoid confusion. Here, experimental group for first post-test (catheter not yet placed) t is 12.443, and for control group with second post-test (catheter placed) t is 12.308. How to justify these two very close values for different settings?
- Authors are advised to restructure the long sentences into smaller ones with clear meaning.
Overall, the paper contains interesting and revealing content. Authors are advised for minor revision of this paper.
Comments on the Quality of English Language
Authors are advised to restructure the long sentences into smaller ones with clear meaning to avoid any ambiguity.
Response:
#Thanks to the reviewer for his insightful suggestions, the authors have revised the Excel in the attached document lines 77, 79, and 94 to MS-Excel.
#The definition and description of CVI are in the attachment lines 131-136.
#The authors have corrected the SPSS for Windows 25.0 in line 203 in the attachment to SPSS 25.0 for Windows.
#The conditions of the study participants and the estimated sample size are described in the attachment lines 91-100.
#Table 7 has adjusted spacing as shown in lines 369-370 in the attachment.

Reviewer 4 Report
The authors touched on a very crucial topic in today's health care system. Here are my comments for the authors to improve the manuscript.
Major issues:
The paper lack some critical steps and details in the study design and statistical analysis.
1.Page 3, lines 98-101, the author used an independent-sample t-test and a chi-square test to examine the differences between the basic information of the two groups. It is better to provide a brief description (e.g. the data sources and data types) of the basic info of the subjects in the two groups. There was a lack of the results of t-test and more details about the chi-square test should be complemented, such as the steps to calculate the degrees of freedom and p-value. To better review the manuscript, more detailed steps for the other tests used should be included in the text.
2.Statistical comparisons between the experiment and control groups (after catheter placed) are needed.
3. I am not sure how to understand the term ‘AI’ in AI image e-book guidance. There seems no comprehensive theory and methodology of the term ‘AI’ indicated in the manuscript. If the image e-book guidance without AI exists, would it be possible to use the image e-book guidance in the health guidance of CRBD. How AI could facilitate the image e-book guidance? Would it be possible to compare the image e-book guidance with the text paper guidance?
Minor issues:
1.There seems a mismatch of the number ‘1.21’ in line 269 vs the number ‘1.12’ in Table 2.
2.For Tables 1-7, the calculation of ‘Difference in average’ was not clearly described and precise column names and explanation for numerical values should be indicated. Both mean and SD in the tables should have the same number of decimal places. The numbers and lines of Table 7 were somewhat confusing, and certain column names (e.g., "t" and "p") lacked clarity in their meaning. The details about the statistical analysis should be added in the legend note.
3.While writing p values of statistically significant data, instead of p=000, the actual level of significance should be recorded. If p value is smaller than 0.0001, then it can be written as p<0.001.
Author Response
Response to Reviewer Comments
Comments and Suggestions for Authors
The authors touched on a very crucial topic in today's health care system. Here are my comments for the authors to improve the manuscript.
Major issues:
The paper lack some critical steps and details in the study design and statistical analysis.
1.Page 3, lines 98-101, the author used an independent-sample t-test and a chi-square test to examine the differences between the basic information of the two groups. It is better to provide a brief description (e.g. the data sources and data types) of the basic info of the subjects in the two groups. There was a lack of the results of t-test and more details about the chi-square test should be complemented, such as the steps to calculate the degrees of freedom and p-value. To better review the manuscript, more detailed steps for the other tests used should be included in the text.
2.Statistical comparisons between the experiment and control groups (after catheter placed) are needed.
- I am not sure how to understand the term ‘AI’ in AI image e-book guidance. There seems no comprehensive theory and methodology of the term ‘AI’ indicated in the manuscript. If the image e-book guidance without AI exists, would it be possible to use the image e-book guidance in the health guidance of CRBD. How AI could facilitate the image e-book guidance? Would it be possible to compare the image e-book guidance with the text paper guidance?
Minor issues:
1.There seems a mismatch of the number ‘1.21’ in line 269 vs the number ‘1.12’ in Table 2.
2.For Tables 1-7, the calculation of ‘Difference in average’ was not clearly described and precise column names and explanation for numerical values should be indicated. Both mean and SD in the tables should have the same number of decimal places. The numbers and lines of Table 7 were somewhat confusing, and certain column names (e.g., "t" and "p") lacked clarity in their meaning. The details about the statistical analysis should be added in the legend note.
3.While writing p values of statistically significant data, instead of p=000, the actual level of significance should be recorded. If p value is smaller than 0.0001, then it can be written as p<0.001.
Response:
#Thanks to the reviewer for his insightful suggestions, the authors have revised the description of the pre-intervention difference analysis between the experimental group and the control group as shown in the attached file lines 100-106.
#Line 277 in the attachment has been corrected to be the same as Table 2. Pre-test 1.12.
#The attached statistics p=0.000 all mean that the p-value is less than 0.0001, and they are all rewritten as p<0.001.

Round 2
Reviewer 4 Report
The authors addressed my comments.